# Mortality, Morbidity and Health-Related Outcomes in Informal Caregivers Compared to Non-Caregivers: A Systematic Review

**DOI:** 10.3390/ijerph19105864

**Published:** 2022-05-11

**Authors:** Patrick Janson, Kristina Willeke, Lisa Zaibert, Andrea Budnick, Anne Berghöfer, Sarah Kittel-Schneider, Peter U. Heuschmann, Andreas Zapf, Manfred Wildner, Carolin Stupp, Thomas Keil

**Affiliations:** 1State Institute of Health, Bavarian Health and Food Safety Authority, 91058 Erlangen, Germany; kristina.willeke@lgl.bayern.de (K.W.); lisa.zaibert@lgl.bayern.de (L.Z.); manfred.wildner@lgl.bayern.de (M.W.); carolin.stupp@lgl.bayern.de (C.S.); thomas.keil@lgl.bayern.de (T.K.); 2Institute of Clinical Epidemiology and Biometry, University of Würzburg, 97080 Würzburg, Germany; e_heuschma_p@ukw.de; 3Institute of Medical Sociology and Rehabilitation Science, Charité—Universitätsmedizin Berlin, 10098 Berlin, Germany; andrea.budnick@charite.de; 4Institute of Social Medicine, Epidemiology and Health Economics, Charité—Universitätsmedizin Berlin, 10098 Berlin, Germany; anne.berghoefer@charite.de; 5Department of Psychiatry, Psychotherapy and Psychosomatic Medicine, University Hospital Würzburg, 97080 Würzburg, Germany; kittel_s@ukw.de; 6Clinical Trial Center Würzburg, University Hospital Würzburg, 97080 Würzburg, Germany; 7Bavarian State Ministry for the Environment and Consumer Protection, 81925 Munich, Germany; andreas.zapf@stmuv.bayern.de; 8Pettenkofer School of Public Health, University of Munich, 81377 Munich, Germany

**Keywords:** cohort studies, longitudinal studies, cross-sectional studies, family caregivers, informal caregiving, mental health, physical health, population-based studies, systematic review

## Abstract

A systematic overview of mental and physical disorders of informal caregivers based on population-based studies with good methodological quality is lacking. Therefore, our aim was to systematically summarize mortality, incidence, and prevalence estimates of chronic diseases in informal caregivers compared to non-caregivers. Following PRISMA recommendations, we searched major healthcare databases (CINAHL, MEDLINE and Web of Science) systematically for relevant studies published in the last 10 years (without language restrictions) (PROSPERO registration number: CRD42020200314). We included only observational cross-sectional and cohort studies with low risk of bias (risk scores 0–2 out of max 8) that reported the prevalence, incidence, odds ratio (OR), hazard ratio (HR), mean- or sum-scores for health-related outcomes in informal caregivers and non-caregivers. For a thorough methodological quality assessment, we used a validated checklist. The synthesis of the results was conducted by grouping outcomes. We included 22 studies, which came predominately from the USA and Europe. Informal caregivers had a significantly lower mortality than non-caregivers. Regarding chronic morbidity outcomes, the results from a large longitudinal German health-insurance evaluation showed increased and statistically significant incidences of severe stress, adjustment disorders, depression, diseases of the spine and pain conditions among informal caregivers compared to non-caregivers. In cross-sectional evaluations, informal caregiving seemed to be associated with a higher occurrence of depression and of anxiety (ranging from 4 to 51% and 2 to 38%, respectively), pain, hypertension, diabetes and reduced quality of life. Results from our systematic review suggest that informal caregiving may be associated with several mental and physical disorders. However, these results need to be interpreted with caution, as the cross-sectional studies cannot determine temporal relationships. The lower mortality rates compared to non-caregivers may be due to a healthy-carer bias in longitudinal observational studies; however, these and other potential benefits of informal caregiving deserve further attention by researchers.

## 1. Introduction

Ageing populations are a challenge to many societies. In 2019, about 700 million people were aged 65 or older. By 2050, this number is estimated to double to 1.5 billion [1]. The strongest increase will be among the “oldest old”, i.e., those who are most likely to have physical, cognitive, or other functional limitations that may require care [2].

Family members and/or friends are the most important resource for caregiving. The term ‘informal caregivers’ is defined as people who provide any kind of help to older family members, friends, and people in their social network, who live either inside or outside the household of the care recipient. In many European countries, among people aged 50 years or older, almost 20% were informal caregivers in 2017 [3]. Thus, informal caregiving and its potential health effects have become a relevant public health issue [4].

A much-quoted landmark longitudinal study showed strained spousal caregivers had a 63% higher mortality risk than non-caregivers, whereas spousal caregivers who did not feel strained had no elevated adjusted mortality risk [5]. A review almost 20 years ago suggested that among informal caregivers, subjective well-being, physical health, and self-efficacy was reduced as well as stress levels and depression increased [6].

Contrary to previously suggested negative health effects, O’Reilly et al. reported an overall reduced mortality risk and less limiting long-term illness for informal caregivers compared with non-caregivers [7]. Potential positive effects of informal caregiving for relatives with stroke or dementia such as an increased self-esteem, strengthened relationships and lower depressive symptoms have also been reported [8,9].

There have been attempts to explain the partly inconsistent previous findings in population-based studies [10]. Previous research focused on informal caregiving for dementia patients and gender aspects including the situation of married caregivers who may be rather more affected by observing the deterioration of a spouse’s or family member’s health status than by physical strains of caregiving [11].

However, previous analyses often used cross-sectional designs where it is not possible to determine temporal relationships to estimate the incidence of disease. They were often lacking an appropriate comparison group (i.e., non-caregivers from population-based samples), and showed inconsistent findings [5,7,8,9]. Systematic reviews with and without meta-analyses were conducted more than 15 years ago and included studies with nonrepresentative samples [6,12]. The most recent systematic review that we identified included studies until April 2017 and only those that were published in English. Furthermore, that review did not assess mortality as an outcome. Due to more narrowly defined inclusion criteria, they merely screened 666 records after eliminating duplicates compared to the 5513 that we screened [13].

Therefore, we aimed to conduct a global systematic review examining mortality, incidence, and prevalence of mental and physical morbidity as well as health-related quality of life among informal caregivers around the world and to synthesize the results by grouping the outcomes visually differentiated by investigation (longitudinal or cross-sectional).

## 2. Materials and Methods

We searched systematically for relevant studies in three major healthcare databases up to 29 March 2020: CINAHL (via Ebsco Host), MEDLINE (via PubMed) and Web of Science. To consider recent developments in dynamic health care systems, we limited our search to publications of the last 10 years without any language restrictions. We developed protocols for bibliographic searches, inclusion and exclusion criteria, and data extraction before starting the systematic search. We registered this systematic review a priori with PROSPERO (registration number: CRD42020200314).

Figure 1 depicts the selection process of studies on mortality, morbidity and health-related outcomes of informal caregivers compared to non-caregivers. For the reporting, we followed the recommendations of PRISMA (Preferred Reporting Items for systematic Reviews and Meta-Analyses) [14].

### 2.1. Search Strategy

A comprehensive search was conducted using the MeSH term “home nursing” in both databases CINAHL and MEDLINE (using PubMed). Web of Science database does not use terms such as MeSH; therefore, we considered the following search items: (1) “informal caregiver*” OR “home nursing” OR “nursing relatives” OR “caring relatives” OR “caregiving relatives” OR “family caregivers”, (2) “disease*” OR “illness” OR “burden” OR “strain” OR “health” OR “satisfaction” OR “quality of life” OR “effect”. The results of both (1) and (2) were combined by the Boolean operator “AND”. Exclusion items were, “intervention”, “child*”and “palliative” with the Boolean operator “NOT” (Appendix A). Additionally, we screened reference lists of all included articles.

### 2.2. Eligibility Criteria

An informal caregiver was defined as an adult who provides any type of unpaid, ongoing assistance with activities of daily living (ADLs) or instrumental activities of daily living (IADLs) to a family member, friend or neighbour with a chronic illness, disability, or frailty because of age, regardless of the amount of time and duration. We generated a checklist of the inclusion and exclusion criteria in order to document decisions regarding the exclusion of articles.

Eligible studies had to fulfil the following three inclusion criteria: (i) primary population-based study or systematic review (with or without meta-analysis) of population-based studies, or an evaluation of register data or a health insurance database, (ii) comparisons of health related outcomes of informal caregivers (IC) with non-caregivers (NC), and (iii) results reported either as prevalence, incidence, odds ratio (OR), hazard ratio (HR) or a comparable mean- or sum-score of an assessment tool for health related outcomes for both groups.

The exclusion criteria were defined as follows: (i) intervention study; (ii) qualitative study; (iii) narrative review, editorial, or letter; (iv) case study or case series; (v) book (chapter); and (vi) studies with a high or moderate risk of bias, i.e., scores of 3 or more of the max 8-point score (based on Hoy et al.) [15].

### 2.3. Selection of Articles and Study Quality Assessment

One reviewer screened titles and abstracts of identified publications and removed duplicates (PJ). Subsequently, three researchers (PJ, KW, LZ) independently performed the full-text screening; disagreements were discussed and resolved by consensus. In case of disagreement, a fourth reviewer was consulted (TK).

Two reviewers (PJ, KW) assessed the study quality independently, using a 10-item checklist for prevalence studies including a summary score consisting of internal and external validity of the study (Figure 2). External validity was assessed with questions concerning representativeness, sampling, and random selection. The internal validity was assessed with questions concerning nonresponse bias, data collection, case definition, reliability/validity of tool, method of data collection, numerator(s), and denominator(s) [15]. For studies not reporting prevalence rates, the last question of the tool was removed, and the summary score adjusted as follows: low risk of bias = 0–2, moderate risk of bias = 3–5, high risk of bias = 6–8. Disagreements were discussed and resolved by consensus. Relevant data were extracted from the articles, and two tables were composed to summarise basic study information on study design, sample size, mean age and sex, differentiated between longitudinal (Table 1) and cross-sectional studies (Table 2). Table 3 (longitudinal evaluations) and 4 (cross-sectional evaluation) present the assessment tools and outcomes. We allocated variables that were collected only at one time point (e.g., only at baseline) within longitudinal studies to cross-sectional evaluation. Two researchers (KW, LZ) also checked the contents of the tables independently.

## 3. Results

We identified 5531 articles via the search of databases plus 52 studies by hand searching the reference lists of all included articles. After the title and abstract screening and the removal of 18 duplicates, we assessed the full texts of 237 studies for eligibility and regarding inclusion and exclusion criteria. After this process, 22 studies were included in the present review.

### 3.1. Study Design and Setting

Of the 22 included studies, 9 had a longitudinal [16,17,18,19,20,21,22,23,24] and 13 a cross-sectional design [25,26,27,28,29,30,31,32,33,34,35,36,37]. All studies recruited study participants from the general population, except one, which evaluated members of a large insurance company [16]. The follow-up of the longitudinal studies ranged from three to 13 years [19,21]. Eight studies were conducted in Europe [16,18,21,23,30,31,32,33], seven in the USA [17,19,20,24,26,35,36], three in Australia [22,25,37] and Asia [27,28,34], and one in South America [29]. Table 1 and Table 2 summarize the basic characteristic of longitudinal studies and of cross-sectional studies, respectively.

### 3.2. Study Sample

The type and amount of caregiving as well as the relationship with the care recipient and their diseases were rather heterogeneous across the included studies. Some defined informal caregiving as exclusively spousal caregiving [18,22,24,27] or included only women [19,20], whereas others defined an informal caregiver as any family member, relative, or friend [17,21,23,30,32,33]. One study made restrictions according to the amount of care (i.e., ≥5 h of care per week) [37]. Most studies did not define the content of informal caregiving precisely, often using terms such as “any kind of care”. Fredman et al. [19] and Chan et al. [34] classified the content of caregiving with any assistance in activities of daily living (ADL), which is a commonly used approach for assessing someone’s level of functioning in performing everyday tasks but covers a wide range of support (Table 1 and Table 2).

Most of the studies did not distinguish between different types of illnesses of the care recipient. Four studies specifically included informal caregivers who cared for patients with any dementia or only Alzheimer’s disease [27,28,29,36], two studies included only caregivers of cancer patients [25,26], and one study included only caregivers of schizophrenia patients [31] (Table 2).

Sample sizes of the included studies ranged from 114 to 1,122,779 participants [21,36] (Table 1 and Table 2). Fourteen studies assessed depression as an outcome [16,17,18,26,27,28,29,30,31,32,34,35,36,37], eleven health-related quality of life [18,22,25,28,29,30,31,33,34,36,37], and five examined mortality [17,19,21,23,24], anxiety [26,28,29,31,37], or diabetes [17,20,27,28,29] (Table 3 and Table 4).

### 3.3. Longitudinal Evaluations

#### 3.3.1. Mortality

Five out of nine longitudinal studies assessed all-cause mortality [17,19,21,23,24]. All of them showed a lower mortality for informal caregivers compared to non-caregivers (according to varying definitions of the recruited study populations, Table 1) during follow-up periods that ranged from 3 to 13 years [19,21]. The reported hazard ratios for mortality ranged from 0.62 [21] to 0.92 [24] and were all statistically significant although often not sufficiently adjusted for potential confounders (Table 3 and Figure 3).

#### 3.3.2. Mental Morbidities

The analysis of a large German health insurance dataset showed that informal caregivers were more likely to develop mental and behavioural disorders over a five-year period compared with non-caregivers, especially severe stress and adjustment disorders (OR 1.61) and depression (OR 1.38). Furthermore, this study showed an increasing chance for informal caregivers receiving a diagnosis of insomnia (OR 1.22) over a five-year period compared to non-caregivers [16]. This confirmed findings from a large European survey in which informal caregiving was associated with a 29% increase in depressive symptoms in the first two years of providing informal care [18] (Table 3 and Figure 3). 

#### 3.3.3. Physical Morbidities

Rothgang et al. showed that informal caregivers in Germany were more likely to develop musculoskeletal disorders over a five-year period, especially diseases of the spine and back compared to non-caregivers. Furthermore, informal caregiving was associated with higher incidences of joint diseases, various pain conditions and, to a smaller extent, digestive disorders [16] (Table 3 and Figure 3).

#### 3.3.4. Health-Related Outcomes

The European Survey on Health, Aging and Retirement showed, for both male and female informal caregivers, a significantly reduced self-rated health status after two years, but no effect after four and seven years [18]. An Australian study demonstrated a considerable deterioration with borderline statistical significance in the mental QoL component of female high-intensity (≥20 h/week) informal caregivers over a two-year period. There was no difference for male high-intensity informal caregivers compared to non-caregivers. Regarding the physical QoL component, female high-intensity informal caregivers reported a significant decline over two years, whereas male high-intensity informal caregivers showed a significant improvement [22] (Table 3 and Figure 4).

A study in USA examined physical functioning and potential decline by performance-based measures in female informal caregivers aged 65+ years over a six-year period. Low-frequency caregivers showed significantly higher grip strength on average over the follow-up period compared to high-frequency and non-caregivers. There were no differences in the average walking speed or number of chair stands between caregiver groups [20] (Table 3 and Figure 4).

Regarding the use of medical services, a large European multicentre survey reported more doctor visits for informal caregivers compared to non-caregivers at baseline, after 2, 4, and 7 years: 7.6, 8.8, 8.6, 8.0 doctor visits per year (informal caregivers) vs. 5.2, 5.6, 6.3, 6.5 doctor visits per year (non-caregivers). Female informal caregivers visited doctors more often than males [18] (Table 3 and Figure 4).

### 3.4. Cross-Sectional Evaluations

#### 3.4.1. Mental Morbidities

Almost all cross-sectional investigations assessed depression, mostly based on validated questionnaires or DSM IV (Table 2) [17,26,27,28,29,30,31,32,34,35,36,37]. Informal caregivers were more affected by depression than non-caregivers in all studies except one that found no considerable difference [35]. Prevalence estimates of depression ranged from 4% to 51% for informal caregivers compared to 3% to 39% for non-caregivers [26,37]. Studies from Asia showed generally lower prevalences of self-reported doctor’s diagnosis of depression than from other continents. Informal caregiving was strongly associated with depression: ORs ranged from 1.52 [37] to 2.36 [34] (Table 4 and Figure 3).

All five studies examining anxiety (either by self-reported doctor’s diagnosis or validated questionnaires, Table 2) found significantly higher prevalence estimates for informal caregivers compared to non-caregivers [26,28,29,31,37]. For informal caregivers, the prevalence ranged from 2% to 38% compared to 0.8% to 24% for non-caregivers [28,31] (Table 4 and Figure 3).

Based on self-reported doctor’s diagnosis, informal caregivers showed overall higher prevalence rates for insomnia, ranging from 10% to 32% compared to 4% to 19% for non-caregivers [28,29,31]. Gupta et al. reported on the highest prevalence rates but solely included informal caregivers of schizophrenia patients [31] (Table 4 and Figure 3).

#### 3.4.2. Physical Morbidities

Of the five studies examining self-reported doctor’s diagnosis of diabetes [17,20,27,28,29], four showed a higher prevalence among informal caregivers [20,27,28,29], whereas one found no difference [17]. The diabetes prevalence estimates ranged from 6% [28] to 14% [27] (Table 4 and Figure 3).

Overall, informal caregivers had a higher prevalence of self-reported general pain compared to non-caregivers [28,29,31]. Caring for people with schizophrenia was associated with the highest prevalence rate (40%) compared to caring for people with dementia (16%) [28,31] (Table 4 and Figure 3).

Two of the four studies assessing self-reported diagnosis of hypertension showed a higher prevalence for informal caregivers compared to non-caregivers, ranging from 18% [28] to 23% [29] for informal caregivers. The other two studies reported similar prevalences for informal caregivers and non-caregivers, both on a relatively high level: 33% vs. 32% [27] and 57% vs. 58% [17] (Table 4 and Figure 3).

#### 3.4.3. Health-Related Outcomes

The Short Form Survey (SF, different versions) was the most common tool to assess mental and physical components of quality of life [25,28,29,31]. Eight studies showed a significantly worse health-related quality of life among informal caregivers compared with non-caregivers for all measured components [25,28,29,30,31,33,34,37], whereas one study found no difference [36]. Four studies assessed self-rated health [30,34,36] or subjective wellbeing [33] (Table 4 and Figure 4).

Two of three studies showed significantly more self-reported outpatient visits in the past 6 months for informal caregivers for their own health problems compared with non-caregivers [28,29], whereas one study assessing only the last month showed no such difference [34]. Self-reported doctor visits by informal caregivers ranged from 1.1 [29] to 1.3 per month [28] (Table 4 and Figure 4).

## 4. Discussion

### 4.1. Main Findings

The majority of the 22 population-based studies with good quality that we identified for our systematic review on mortality and morbidity of informal caregivers was conducted in Europe and in the USA. Only few studies came from Asia, Australia, and South America.

Results from the longitudinal studies showed a lower total mortality of informal caregivers compared to non-caregivers. Regarding the incidence of chronic diseases, on the contrary, the longitudinal evaluation of a large German health insurance company’s database showed increased risks of developing chronic disorders over a five-year period among informal caregivers. These included especially severe stress and adjustment disorders, depression, diseases of the spine and back, and pain conditions.

Results from most cross-sectional studies supported findings from the longitudinal studies on chronic morbidity. They found higher prevalence estimates for informal caregivers especially for depression, anxiety, sleep disorders, and pain compared to non-caregivers. Furthermore, in cross-sectional investigations, informal caregiving was associated with a reduced mental as well as physical health-related quality of life.

### 4.2. Interpretation of the Results

#### 4.2.1. Comparison with Other Systematic Reviews

Although previous systematic reviews and meta-analyses included important study designs, longitudinal and cross-sectional studies, the authors did not strictly differentiate their results and conclusions taking the considerable differences of the design into account [6,12,13]. The previous reviews and meta-analyses examined mental and physical chronic disorders of informal caregivers compared to non-caregivers but did not examine the mortality of the informal caregivers [6,12,13]. Mortality, as the most relevant endpoint in longitudinal studies, should be included to assess the full impact of a potentially fatal exposure that has been associated with depression, cardiovascular and other relevant chronic morbidity [38]. Previous reviews suggested a negative impact of informal caregiving on mental health such as an increased risk of depression and stress, whereas the effects on physical health were smaller and less consistent [6,12,13]. With an updated literature search and stringent study quality assessment, our systematic review seems to confirm previous results in terms of chronic morbidity. However, the limited longitudinal evidence due to a lack of good-quality cohort studies does not allow to make strong conclusions on temporal relationships with informal caregiving and most chronic diseases. In terms of mortality, our systematic review found five longitudinal studies showing that informal caregivers had a lower mortality risk compared to non-caregivers. We found no study that indicated an increased mortality risk. Although this is strong evidence, the identified studies cannot explain the mechanism of this association. These and other potentially positive aspects of informal caregiving deserve more attention by health scientists and researchers in order to draw a broader picture.

#### 4.2.2. Longitudinal Evaluations

##### Mortality

The results from the longitudinal studies in our comprehensive systematic review showed that informal caregivers seemed to have a lower mortality compared to non-caregivers [17,19,21,23,24]. They confirmed similar findings by O’Reilly et al. [7] and may be explained by a “healthy informal caregiver bias” among the participants of longitudinal studies. Unhealthy family caregivers with a higher mortality risk may be more likely to drop out during the course of the study than healthy caregivers. Those with health problems in general might be less likely to undertake caregiving tasks and to become an informal caregiver because of their own health limitations. Then, without imputations of missing data from participants who dropped out, their outcome assessments remained incomplete. There have been some longitudinal studies that suggest that informal caregivers are healthier [39,40,41]: the study from Bertrand et al. showed that cognitive outcomes in older women caregivers were better than in non-caregivers of the same age [40]. McCann et al. showed that caregiving older adults were physically healthier than their non-caregiving counterparts, and Fredman et al. showed a lower rate of functional decline among informal caregivers [40,41].

Furthermore, the included longitudinal studies assessed the caregiving status only at baseline but did not consider transitions out of or into caregiving. None of the included studies recruited only non-caregivers and separated possible informal and non-caregivers during the course of the study.

Some studies looked at factors determining predictors for mortality. Fredman et al. showed that low-stress caregivers had a 33% lower mortality risk than low-stress non-caregivers did [42]. This result was also in line with Schulz and Beach’s findings, where strained spousal caregivers had a 63% higher mortality risk than non-caregivers, whereas spousal caregivers who were not strained had no elevated mortality risk [5]. More hours of care per week was also associated with an increased mortality risk in two studies in UK [21,23]. An explanation for this effect was suggested by a study from USA showing that more caregiving hours were associated with a biomarker of cellular aging, a shorter relative telomere length, which has been linked with earlier mortality and higher disease risk [43].

##### Morbidity

Only two studies assessed morbidity in a longitudinal study design and showed associations between informal caregiving and depression [16], even if only in the short term [18]. The results are in line with a population-based study of informal caregiving adults aged 40 and older in Germany, demonstrating that informal caregiving affects mental health [44]. Rothgang et al. also showed that the amount of care was associated with the informal caregiver’s health. Caregivers who did not provide care daily had depression less often than those who provided care daily [16]. Regarding physical health, this study reported higher odds ratios for a number of specific diseases at baseline and higher odds ratios to develop a disease within the following five years. These diseases according to ICD-10 diagnosis included, among other things, musculoskeletal disorders including back pain and joint disease. The results showed that informal caregivers were sicker at the beginning and became sicker compared to non-caregivers [16].

##### Health-Related Outcomes

In one cohort study, informal caregivers showed lower mean scores of self-reported health after two years [18]. This is consistent with the results of another cohort study, which found a statistically significant deterioration of the quality of life for high-intensity informal caregivers compared to non-caregivers after two years, but only in females [22]. One explanation for this sex-specific effect may be a reporting bias; however, it remains unclear whether female informal caregivers are more vulnerable or would more often report higher burden of informal caregiving compared to males [45]. Regarding self-reported health, this impact seems to be a short-term effect and is no longer observable after four and seven years [18]. One explanation may be that the study on self-reported health used data from a large European Survey including 11 European countries [18]. The impact of informal caregiving may have been confounded by country-specific differences with respect to preventive and supportive structures for informal caregivers [33]. Furthermore, as in other observational longitudinal studies, selective attrition could have biased the results. The effects of informal caregiving were smaller and insignificant for the subsample that participated in all waves [18].

Physical health-related outcomes seem to be less affected less by informal caregiving compared to mental health. This may be explained by the fact that the maintenance of physical health and function is crucial to the ability of older adults to start and continue caregiving [40]. Rosso et al. reported on a higher grip strength for low-frequency informal caregivers at baseline and after six years of follow-up but did not control for baseline grip-strength [20]. These results are in line with the study of Fredman et al., who measured change in physical functioning over a two-year period and found that informal caregivers deteriorated less compared to non-caregivers—but did not match the two groups on health status at baseline [41].

#### 4.2.3. Cross-Sectional Evaluations

##### Morbidity

Our finding that informal caregiving was associated with negative effects on mental health and especially depression confirmed earlier studies [6,13]. We found that the results for anxiety were similar as for depression, regardless of the applied assessment tool. This confirmed the results of the LASER-AD study on persons with Alzheimer’s showing that most depressed caregivers also reported symptoms of anxiety [46]. Relatives caring for persons with schizophrenia in particular more often reported feelings of shame, resentment, and grief than other caregivers [47]. Informal caregiving, particularly for dementia and schizophrenia patients, was associated with considerably higher prevalences of depression [28,29,31,36]. Caring for people with dementia may cause stress resulting from their cognitive deterioration. This may lead to a unidirectional relationship, the inability to communicate with the care recipient, the challenge of coping with new behavioural problems, and feelings of loss and loneliness, a state called “relational deprivation” [48,49,50]. It is recognized that there is an association between stress and depression [51,52]. The interaction between chronic psychosocial stress, the hypothalamic–pituitary–adrenocortical (HPA) axis, and depression is complex, with inconsistent results in previous studies. It remains unclear whether chronic psychosocial stress leads to alterations in the HPA axis and thereby increases the risk of depression or if individuals with a genetically altered HPA axis reactivity are at increased risk of developing depression if they are exposed to chronic psychosocial stress. Additionally, depression itself can be perceived as a major stressor and thereby lead to alterations in the HPA axis [53,54].

Caring for a person with schizophrenia may be burdensome due to feelings of shame and stigmatisation, resentment, grief, and maladaptive behaviour of the patient and may lead to considerable distress in the family carer [47,55]. The persistence of such stress may lead to anxiety disorders [56]. A present systematic review and meta-analysis found a strong link between subjective burden and anxiety. The results suggested that subjective burden is a crucial determinant of anxiety-related distress in informal caregiving [57].

##### Health-Related Outcomes

Informal caregiving was associated with reduced QoL in self-reported health [25,28,29,30,31,33,34,37], except in one study [36]. These associations underline the results from longitudinal studies [18,22] and are also in line with findings from another cross-sectional study in Germany [58], with international studies [59,60] as well as a meta-analysis who assessed well-being [6].

### 4.3. Positive Aspects of Informal Caregiving

Apart from a potentially reduced mortality, our systematic review did not identify much evidence for protective effects of informal caregiving in terms of reduced chronic morbidity, although we did not exclude these outcomes. Beyond the focus of our present work based on stringent methodological quality assessments, however, further research exists suggesting that the provision of emotional and practical support to others may result in improved mental and/or physical health for the provider of such support [61,62,63]. Informal caregiving can lead to a high level of self-esteem and a positive change in the sense of mastery among female informal caregivers when caring for a non-resident care recipient [64,65]. In addition, it also seems to have a stress-buffering effect, which leads to lower mortality [17].

Informal caregivers constitute a heterogeneous group regarding the amount of care, the caregiving situation (resident or non-resident, caring for a parent or a spouse), the disease of the care recipient, and perceived social support. All of these factors can contribute to experiencing a situation as stressful, which may be experienced as eustress (i.e., beneficial stress) but often as distress. According to the definition of Lazarus and Folkman, this occurs if a person appraises the (caregiving) situation as taxing or exceeding their available resources [66].

### 4.4. Strengths and Limitations

Among the strengths of our systematic review and difference to previous reviews was the stringent formal evaluation of the methodological quality of all eligible studies using an established assessment tool. Studies with lower quality, i.e., a high or moderate risk of bias, were not included in the present review [15]. Furthermore, we excluded selective and convenience samples and focused only on studies with samples from the general population to be able to judge the generalizability of the results with respect to the source population. Another strength was our comprehensive literature search in major databases without language restrictions and a complementary hand search, thus trying to take a global perspective on the disease burden of informal caregiving. We also considered a broad range of outcomes and assessment instruments, which allowed us to assess different morbidity and health-related outcomes.

However, several limitations of our systematic review must be noted. First, most of the investigations that we included were cross-sectional analyses. They did not allow assessing the causal direction of informal caregiving and outcomes. A chronic disease or condition may have existed before the provision of informal care started. Only longitudinal studies can assess temporal relations. However, we were only able to identify a few. In the present review, results for morbidity outcomes that were examined in both study types were rather consistent. Second, varying caregiving arrangements or situations hamper comparisons of study results. Some studies recruited only female caregivers or spousal caregivers, whereas others included persons regardless of their relationship with the index patient. Most of the studies did not assess an individual appraisal of the informal caregiver regarding their care situation as well as motivations and willingness. Having no choice in becoming an informal caregiver is associated with negative health effects including higher levels of emotional stress and physical strain [67]. Buyck et al. showed that the individual appraisal of the caregiving situation matters with respect to health-related outcomes. Informal caregivers who experience their situation as a burden showed significantly worse physical and mental health compared to non-caregivers. Interestingly, informal caregivers who showed low levels of burden reported better perceived health and less depressive symptoms compared to non-caregivers (OR: 0.50, 95% CI 0.37–0.68) [68]. With respect to the care recipient, the situation was similar. Many studies included only patients with a specific disease (dementia, cancer, or schizophrenia), whereas some studies were less focused and included patients with different diseases. However, although there were varying categories of dyads across the studies, we found rather consistent results for an association with depression and anxiety. This may suggest that the outcomes are to some extent independent of carer or recipient characteristics. Third, although we did not exclude any article because of its language, we may have missed studies in journals that were not referenced by the major medical and health science databases included in our search. However, by searching the Cumulative Index to Nursing and Allied Health Literature (CINAHL), the major database of journal articles about nursing, allied health, biomedicine and healthcare, in addition to MEDLINE and Web of Science as well as hand search, we presume to have identified most relevant population-based studies on informal caregiving. Fourth, our literature search was conducted before the current SARS-CoV-2 pandemic. Therefore, it did not include studies examining the potential effects of non-pharmaceutical interventions due to the pandemic and lockdown measures [69]. It seemed that informal caregivers had to deliver more caregiving hours per week [70] and experienced additional burden, worsened care situations [71], and a significant deterioration of their wellbeing and increased rates of depression due to the pandemic situation compared to non-caregivers [72]. Fifth, most of the studies were conducted in Western, developed countries. Cultural differences regarding informal caregiving which might have an impact on health were not considered. Sixth, differences in healthcare systems may contribute to different health outcomes of family carers. In many countries, informal caregiving is part of the welfare contract between the citizens and the public social and healthcare system. There may be health-system-related and/or cultural differences on how much relatives may feel obliged to become engaged in informal caregiving. However, the studies we included in the present review did not collect data on these complex issues, which require further multidisciplinary research efforts. Seventh, additionally to population-based studies, we also included register data or health insurance databases. Since some health insurance companies in Germany cover a large part of the population and membership is open to all employees, with very few exceptions, we did not want to exclude them from the review. One advantage of using routine care data (retrospectively) from large insurance databases is that they include data from persons who would usually not consent to participate in prospective research projects; thus, this data may even be subject to less selection bias than research projects including original data. However, although no single health insurance company in Germany can be considered as representative for the whole country, data from the members of a large company that include information of the family caregiver status may provide additional insight into potential associations [73].

## 5. Conclusions

For our systematic review, we were able to identify several observational population-based studies with good quality. It seemed that informal caregivers had a lower mortality risk compared to non-caregivers. A healthy carer bias in longitudinal population-based studies may have contributed to this finding, but informal caregiving may have positive causal effects for the health of family caregivers. These potential benefits require further research.

In terms of chronic morbidity, our systematic review showed statistical associations of informal caregiving with the development of severe stress and adjustment disorders, depression, anxiety, sleep disorders, diseases of the spine and back, pain conditions, and a lower quality of life. These effects seemed stronger among informal caregivers who cared for dementia or schizophrenia patients. However, these results need to be interpreted with caution, as most of the included studies had a cross-sectional design, which does not allow to determine temporal or causal relationships.

Future longitudinal observational studies on the disease burden of informal caregivers should start at the beginning of providing informal care and aim to avoid high drop-out rates to better understand potential long-term health effects.

Targeted intervention strategies should aim to prevent severe stress disorders, depression, and anxiety, as well as spinal diseases, and back and pain conditions. Beyond these findings from our study, we suggest that concepts to empower informal caregivers need to be planned in collaboration with formal nursing and care services and adhere to stringent evaluations of sustainable benefits.

## Figures and Tables

**Figure 1 ijerph-19-05864-f001:**
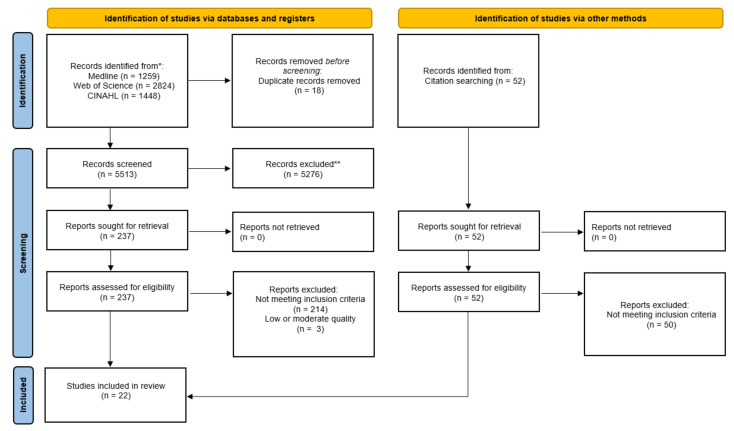
PRISMA 2020 flow diagram showing the selection process of studies on mortality, morbidity and health-related outcomes of informal caregivers compared to non-caregivers.

**Figure 2 ijerph-19-05864-f002:**
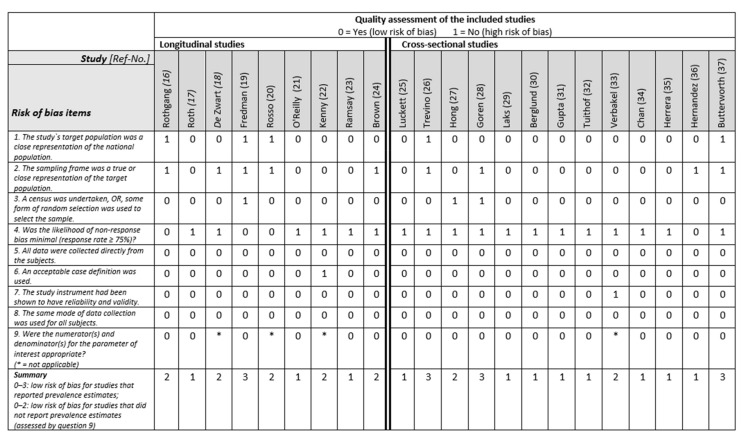
Overview of the quality assessment of the included studies based on the risk of bias criteria as suggested by Hoy et al. [15].

**Figure 3 ijerph-19-05864-f003:**
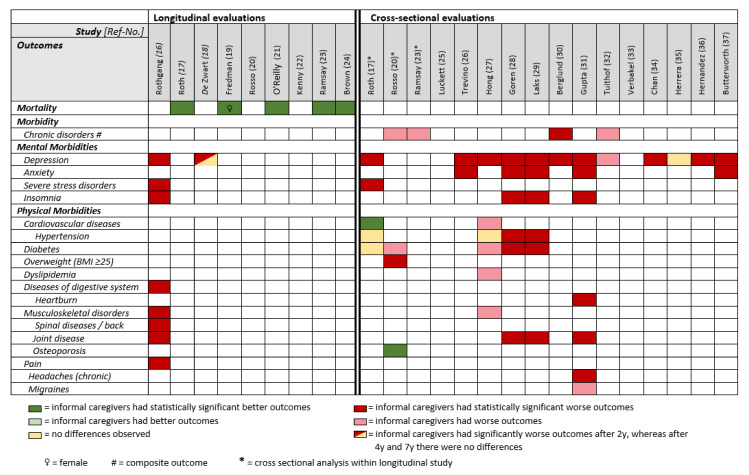
Simplified overview comparing mortality/morbidity outcomes between informal caregivers and non-caregivers in observational studies.

**Figure 4 ijerph-19-05864-f004:**
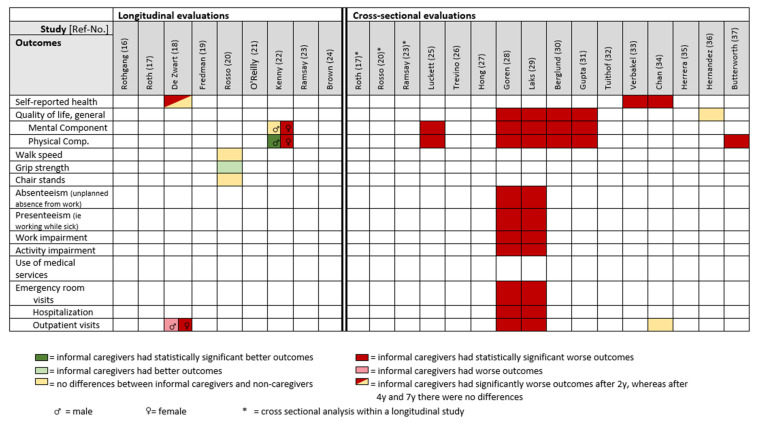
Simplified overview comparing self-reported general health-related outcomes and health care utilization between informal caregivers and non-caregivers in population-based studies.

**Table 1 ijerph-19-05864-t001:** Basic characteristics of longitudinal studies comparing occurrence of diseases/disorders between informal caregivers (IC) and non-caregivers (NC).

First Author, Publication Year, Country [Reference]	Year of Recruitment, Follow-up	Study Population: N, Age [Mean (SD); Range], Women (%)	Data Source	Description of Sample
Rothgang, 2018, Germany [16]	2012, 5 y	IC vs. NC: 179,134 vs. approx. 2,300,000 (projected) members as NC from BARMER health insurance company. IC (for NC not reported) 20–44 y: 11%, 45–99 y: 89% 67%	BARMER health insurance company: routine data of IC (*n* = 179,134). Weighted for general population characteristics.	The main caregiver of the care recipient based on reports to their statutory health insurance company (in Germany). The care recipient had to be in need of care according to the German Social Security Code XI. Main caregiver sociodemographic parameters: male: 30.5%, female: 69.5%; age: 0–49 y: 14.4%, 50–59 y: 26.0%, 60–69 y: 25.7%, 70–79 y: 16.2%, >80 y: 17.7%; working hours per week: no: 65.3%, 0–9 h/wk: 4.7%, 10–19 h/wk: 11.0%, 20–29 h/wk: 10.7%, >30 h/wk: 8.4%.
Roth, 2018, USA [17]	2003, 7 y	IC vs. NC: 3580 vs. 3580, 63.6 y (9.0) vs. 63.6 y (9.1), 63.0% vs. 61.3%	The population-based Reasons for Geographic and Racial Differences in Stroke study.	The caregiver had to be older than 45 years and provide any kind of care to a family member with chronic illness or disability.
De Zwart, 2017, AT/BE/CH/DE/DK/ES/FR/GR/IT/ NL/SE [18]	2004–2013, 7 y	IC vs. NC: 423 vs. 10,048, 68.0 y vs. 61.8 y, n.r.	The Survey of Health, Ageing and Retirement in Europe including people ≥ 50 years and their spouses.	The caregiver had to provide any personal care (i.e., washing, getting out of bed, or dressing) to their partner (spouse) daily or almost daily during at least three months within the past 12 months.
Fredman, 2015, USA [19]	1997, 13 y	IC vs. NC: 374 vs. 694, 81 y (0.2) vs. 81.5 y (0.1), ≥65, 100%	Caregiver-Study of Osteoporotic Fractures (Caregiver-SOF), an ancillary study to the SOF.	The caregiver had to be a woman, assisting someone with at least one activity of daily living (ADL) or instrumental activity of daily living (IADL). Care recipients’ characteristics: 27% dementia, 22% frailty/general health decline, 21% stroke. Years spent caregiving (at baseline of study): <2 y: 22.2%, 2–5 y: 39.6%, >5 y: 38.2%.
Rosso, 2015, USA [20]	1993–1998, 6 y	IC vs. NC: 2138 vs. 3511, 70.1 y (3.8) vs. 69.8 y (3.7), 100%	A subsample from the Women’s Health Initiative Clinical Trial of women aged 65–80 years.	The caregiver had to provide any kind of care for a family member or friend because of being sick, limited or frail.
O’Reilly, 2015, UK (N. Ireland) [21]	2011, 2.8 y	IC (1–19 h/wk, 20–49 h, ≥50 h) vs. NC: 183,842 vs. 938,937, 25–44 y: 35%, 36%, 28% vs. 42%, 45–64 y: 53%, 49%, 44% vs. 35%, ≥65 y: 12%, 16%, 28% vs. 23%, 59% vs. 51%	Northern Ireland Mortality Study based census data and mortality register.	The caregiver had to provide any kind of care to a family member, friend, neighbour or others with long-term physical or mental illness, disability or problems related to old age.
Kenny, 2014, Australia [22]	2001, 4 y	IC vs. NC: 424 vs. 424, 48.9 y (14.8) vs. 49.7 y (15.2), ≥16 y, 60%	IC and propensity score-matched NC from the Household Income and Labour Dynamics.	The caregiver had to provide any kind of care to a disabled spouse, adult relative or elderly parent/parent-in-law. Distribution of caregiving hours/week: <5 h/wk: 49.1%, 5–19 h/wk: 33.5%, >20 h/wk: 17.4.
Ramsay, 2013, UK [23]	2001, 8 y	IC (1–19 h/wk and ≥20 h/wk) vs. NC: 31,404 vs. 146,964, 52.0 y (9.4) and 53.7 y (10.7) vs. 50.7 y (10.9), 55% and 61% vs. 49%	Office for National Statistics-Longitudinal Study of England and Wales (people between 35–74 years).	The caregiver had to look after or give any help/support to family members, friends, neighbours, or others with long-term physical or mental ill health, disability, or age-related problems.
Brown, 2009, USA [24]	1993, 7 y	IC (1–14 h/wk and >14 h/wk) vs. NC: 306 and 338 vs. 2732, range ≥70 y, n.r.	Health and Retirement Study, a nationally representative sample.	The caregiver had to provide any kind of care to their spouse.

n.r. = not reported; y = years; h/wk = hours/week.

**Table 2 ijerph-19-05864-t002:** Basic characteristics of cross-sectional studies comparing occurrence of diseases/disorders be-tween informal caregivers (IC) and non-caregivers (NC).

First Author, Publication Year, Country [Reference]	Year of Recruitment, Follow-Up	Study Population: N, Age (Mean (SD); Range), Women (%)	Data Source	Description of Sample
Luckett, 2019, Australia [25]	2016	IC vs. NC: 374 vs. 1993, 49 y (17) vs. 47 y (20), 59% vs. 48%	The Health Omnibus Survey: annual survey with randomly selected households.	The caregiver had to provide any kind of care for someone with cancer in the last five years.
Trevino, 2018, USA [26]	2002–2008	IC vs. NC: 540 vs. 9282, 53 y (14) vs. 53 y (18), 72% vs. 72%	The Coping with Cancer study identified IC and the National Comorbidity Survey Replication: general population data.	The caregiver had to provide any kind of care for a relative or a friend with advanced cancer (estimated life expectancy of six months or less).
Hong, 2017, South Korea [27]	2012–2013	IC vs. NC: 3868 vs. 310,658, 61 y (14) y vs. 53 y (14) y, 48% vs. 48%	Korea Community Health Survey.	Spousal caregiving for a partner with dementia.
Goren 2016, Japan [28]	2012–2013	IC vs. NC: 1302 vs. 53,758, 53 y (14) vs. 48 y (16), 53% vs. 49%	National Health and Wellness Survey in Japan.	Caring for a related adult with dementia including Alzheimer’s disease.
Laks, 2016, Brazil [29]	2012	IC vs. NC: 209 vs. 10,644, 42 y (14) vs. 40 y (16), 53% vs. 50%	The National Health and Wellness Survey: internet-based survey, using stratified random sampling	Any kind of care for a person with dementia.
Berglund 2015, Sweden [30]	2004–2013	IC vs. NC: 9343 vs. 76,112, 54 y (15) vs. 49 y (18), 59% vs. 54%	Swedish national public health survey Health on equal terms.	Any kind of care for a sick or old relative.
Gupta, 2015, FR/DE/IT/ES/GB [31]	2010, 2010 and 2013	IC vs. NC: 398 vs. 158,989, 45 y (16) vs. 46 y (16), 60% vs. 51%	The 5EU National Health and Wellness Survey: stratified random sample.	Any kind of care to a person with schizophrenia.
Tuithof, 2015, The Netherlands [32]	2010–2012	IC vs. NC: 1759 vs. 3544, <45 y: 36% (21–68 y) vs. 55% (21–68 y), 60% vs. 45%	The 2nd wave of the Netherlands Mental Health Survey and Incidence Study-2: nationally representative sample	Providing unpaid care in the 12 months preceding the study to a family member, partner, or friend because of physical or mental problems, or ageing.
Verbakel, 2014, AT/BE/CZ/DK/FI/FR/DE/HU/IE/LU/NL/NO/PL/SK/SI/ES/SE/GB [33]	2007	IC vs. NC: 4736 vs. 15,600, n.r. vs. n.r. n.r. vs. n.r.	The European Quality of Life Survey: random samples of the adult population; selection of countries based on availability of all relevant data.	Any kind of care for an elderly or disabled relative.
Chan, 2013, Singapore [34]	2010–2011	IC vs. NC: 1077 vs. 318, 56 y (13) vs. 57 y (15), 61% vs. 65%	A stratified, random sample of 20,000 Singaporeans from the national database of dwellings.	Any kind of care for a family member or a friend aged ≥75 y.
Herrera, 2013, USA [35]	1998–1999	IC vs. NC: 92 vs. 1888, 77 y (0.50) vs. 77 y (0.14), 72% vs. 59%	The Hispanic Established Populations for Epidemiologic Studies of the Elderly (Wave 3).	Caring for a related or unrelated older adult. Mexican-American caregivers aged ≥70 y from Texas, New Mexico, Colorado, Arizona, and California.
Hernandez, 2010, USA [36]	2000–2001	IC vs. NC: 57 vs. 57, 78 y (4) vs. 79 y (5), 68% vs. 68%	The Hispanic Established Populations for Epidemiologic Studies of the Elderly (Wave 4).	Mexican American caregivers aged ≥65 y from Texas, New Mexico, Colorado, Arizona, and California. Caring for a person with Alzheimer’s disease or physical disability.
Butterworth, 2010, Australia [37]	2005	IC vs. NC: 212 vs. 2010, 67 y (0.03), 64–69 y vs. 67 y (0.10), 64–69 y, 59% vs. 47%	PATH Through Life Project: survey of 3 cohorts from Canberra and Queanbeyan: second wave data of cohort born 1937–1941 For the present analysis.	Any kind of care ≥5 h per week. The sample of care recipients consisted of: physical disability/chronic illness (58%), memory/cognitive problems (10%), mental illness (13%).

n.r. = not reported; y = years.

**Table 3 ijerph-19-05864-t003:** Longitudinal evaluations: disease burden outcomes among informal caregivers (IC) compared with non-caregivers (NC).

First Author, Publication Year [Reference]	Outcomes and Assessment Tools	Results: Informal Caregivers (IC) vs. Non-Caregivers (NC) (Numbers in Bold Were Reported to Be Stat. Significant Results)
**Rothgang**, 2018 [16]	Incidence based on International Statistical Classification of Diseases and Related Health Problems (ICD-10):	**Five-year incidence of disease in IC vs. NC** (2012–2017; as ref.-categ.) comparing new diagnoses in 2017 with 2012 using odds ratios (OR) as relative risk estimates:
	**1. Mental and behavioural disorders** (F)	**1. OR: 1.35** (prevalence in 2012: 39.6% vs. 36.7%; in 2017: 48.7% vs. 42.5%)
	**1a. Depression** (F32, F33, F34.1)	**1a. OR: 1.38** (prevalence in 2012: 18.1% vs. 16.5%; in 2017: 23.4% vs. 19.7%)
	**1b. Severe stress/adjustment disorders** (F43)	**1b. OR: 1.61** (prevalence in 2012: 8.5% vs. 7.1%; in 2017: 12.5% vs. 8.5%)
	**1c. Sleep disorders** (F51)	**1c. OR: 1.2** (prevalence in 2012: 1.2% vs. 1.1%; in 2017: 1.8% vs. 1.5%)
	**2. Diseases of digestive system** (K)	**2. OR: 1.06** (prevalence in 2012: 39.2% vs. 37.6%; in 2017: 45.9% vs. 44.6%)
	**3. Diseases of musculoskeletal system and connective tissue** (M)	**3. OR: 1.17** (prevalence in 2012: 66.8% vs. 64.4%; in 2017: 72.1% vs. 69.4%)
	**3a. Spinal diseases/back** (M40-54)	**3a. OR: 1.19** (prevalence in 2012: 50.6% vs. 47.6%; in 2017: 54.9% vs. 51.3%)
	**3b. Joint disease** (M00-25)	**3b. OR: 1.09** (prevalence in 2012: 20.3% vs19.6%; in 2017: 23.5% vs. 22.7%)
	**4. Pain** (F45.5, F62.80, G54.6, M25.5, M54, M75.8, M79.6, R52)	**4. OR: 1.19** (prevalence in 2012: 42.9% vs. 39.9%; in 2017: 48.4% vs. 44.6%)
**Roth**, 2018 [17]	**Mortality over 7 years** (death certificates or National Death Index)	**Total sample aHR: 0.84, 95% CI: 0.72–0.97), p = 0.018**
		**Subsamples** (by caregiving groups):
		Spouse caregivers: aHR: 0.96, 95% CI: 0.73–1.25
		High strain caregivers: aHR: 0.73, 95% CI: 0.52–1.03
		Some strain caregivers: aHR: 0.89, 95% CI: 0.71–1.12
		No strain caregivers: aHR: 0.84, 95% CI: 0.66–1.07
		Caregiving ≥ 14 h/wk: **aHR: 0.78, 95% CI: 0.63–0.98**
		Caregiving < 14 h/wk: aHR: 0.87, 95% CI: 0.71–1.07
**De Zwart**, 2017 [18]	1. **Depressive symptoms** **EURO-D scale** (0 = not depressed at all, 12 = severely depressed)	**1. Change in mean scores after propensity match scoring**: males: after 2 y: **0.45 (0.16), *p* < 0.01**, 4 y: −0.18 (0.18), 7 y: 0.15 (0.23), both p ≥ 0.050 females: after 2 y: **0.57 (0.16), *p* < 0.01**, 4 y: −0.10 (0.18), 7 y: −0.13 (0.20), both p ≥ 0.050
	**2. Self-reported health** (5-point scale from 1 = worst to 5 = best)	**2. Change in mean scores after propensity match scoring**: males: after 2 y: **−0.16 (0.07), *p* < 0.10**, 4 y: 0.07 (0.09), 7 y: 0.02 (0.10), both p ≥ 0.050 females: after 2 y: **−0.20 (0.061), *p* < 0.10**, 4 y: 0.01 (0.07), 7 y: 0.02 (0.08), both *p* ≥ 0.050
	**3. Self-reported number of doctor visits in past 12 months**	**3. Change in mean scores after propensity match scoring**: males: after 2 y: 0.67 (0.50), *p* ≥ 0.050, 4 y: 0.88 (0.64), *p* ≥ 0.050, 7 y: 1.22 (0.79), *p* ≥ 0.050 females: after 2 y: **1.37 (0.47), *p* < 0.05**, 4 y: 0.01 (0.52),p ≥ 0.050, 7 y: **−1.54 (0.58), *p* < 0.05**
**Fredman**, 2015 [19]	**Mortality over 13 years** (death certificates)	IC vs. NC: 38.8% (*n* = 145) vs. 48.7% (*n* = 338) deaths **aHR 0.77, 95% CI: 0.62–0.95**
**Rosso**, 2015 [20]	**Physical function**	**Baseline Characteristics for High-Frequency IC (≥3 x/wk)/Low-Frequency IC (≤2 x/wk)/NC**
	**1. Mean walk speed** (time to complete a 6-m course)	**1. Mean walk speed**, m/s (SD): 1.10 (0.26)/1.08 (0.27)/1.09 (0.26)
	**2. Mean grip strength** (by hand-grip dynamometer)	**2. Mean grip strength**, kg (SD): 22.5 (5.5)/23.2 (5.4)/22.9 (5.4)
	**3. Mean chair stands** (number of times participants could rise in 15 s)	**3. Mean chair stands**, number (SD): 6.4 (1.9)/6.4 (1.9)/6.4 (1.9)
	**Mean Differences in Measures of Physical Function after 6 years: High-frequency IC** (≥3 x/wk) and **Low-Frequency** (≤2 x/wk) vs. **NC** (reference)	**1. Walk speed (m/s)**: 0.01, 95% CI: −0.01–0.03 and 0.00, 95% CI: −0.12–0.02
		**2. Grip strength (kg)**: 0.11, 95% CI: −0.57–0.35 and **0.63, 95% CI: 0.24–1.01**
		**3. Chair stands (number)**: 0.02, 95% CI: −0.17–0.22 and −0.12, 95% CI: −0.26–0.03
		Analyses used inverse proportional weights from propensity scores of caregiving at baseline and for differential attrition and were adjusted for study enrolment.
**O’Reilly**, 2015 [21]	**Mortality over 2.8 years** (mortality records)	**Total sample: aHR 0.72, 95% CI: 0.69–0.75**
		**Subsamples** (by number of hours/week spent caring):
		Men, heavy care (≥50 h/wk): **aHR: 0.77, 95% CI: 0.71–0.83**
		Men, medium care (20–49 h/wk): **aHR: 0.81, 95% CI: 0.71–0.92**
		Men, light care (1–19 h/wk): **aHR: 0.70, 95% CI: 0.64–0.77**
		Women, heavy care (≥50 h/wk): **aHR: 0.76, 95% CI: 0.69–0.83**
		Women, medium care 20–49 h/wk): **aHR: 0.66, 95% CI: 0.57–0.78**
		Women, light care (1–19 h/wk): **aHR: 0.62, 95% CI:0.56–0.69**
**Kenny**, 2014 [22]	**Quality of Life (QoL)** (SF-36: Physical Component and Mental Component Scale, range 0 = worst, 100 = best)	Coefficient (95% CI) from separate multiple regression models for change in QoL components in IC relative to NC:
	**1. Physical Functioning Component**	1. Caregiving 5–19 h/wk: after 2 y: 2.5 (−4.8–9.9), after 4 y: −7.7 (−16.4–1.0) 1. Caregiving ≥ 20 h/wk: after 2 y: **10.0 (1.5–18.4)**, after 4 y: 3.1 (−6.7–12.9)
	**2. Mental Health Component**	2. Caregiving 5–19 h/wk: after 2 y: −2.4 (−7.4–2.5), after 4 y: **−9.2 (−17.0–1.5)**, 2. Caregiving ≥ 20 h/wk: after 2 y: 3.2 (−3.5–9.9), after 4 y: −8.7 (−18.1–0.7)
**Ramsay**, 2013 [23]	**Mortality over 8 years**	**All-cause mortality in subsamples**
		Men Caregiving ≥20 h/wk: **aHR 0.87, 95% CI: 0.79–0.97**
		Men Caregiving 1–19 h/wk: **aHR 0.81, 95% CI: 0.75–0.89**
		Women Caregiving ≥20 h/wk: **aHR 0.80, 95% CI: 0.71–0.89**
		Women Caregiving 1–19 h/wk: **aHR 0.74, 95% CI: 0.66–0.83**
**Brown**, 2009 [24]	**Mortality over 7 years**	**All-cause mortality**
		Caregiving ≥14 h/wk: **aHR 0.64, 95% CI: 0.45–0.90**
		Caregiving 1–14 h/wk: aHR 0.92, 95% CI: 0.69–1.24

CI = confidence interval; aHR = adjusted hazard ratio; vs. = versus; wk = week; SD = standard deviation; SF-36 = 36-Item Short Form Survey.

**Table 4 ijerph-19-05864-t004:** Cross-sectional evaluations: disease burden outcomes comparing informal caregivers (IC) and non-caregivers (NC).

First Author, Publication Year [Reference]	Outcomes and Assessment Tools	Results: Informal Caregivers (IC) vs. Non-Caregivers (NC) (Numbers in Bold Were Reported to Be Stat. Significant Results)
**Luckett**, 2019 [25]	**Quality of Life** (SF-12, range 0 = worst, 100 = best)	**Physical component summary** (PCS): mean (SD): **49.1 (10.2)** vs. **50.4 (10.0)**,***p* = 0.020** **Mental component summary** (MCS): mean (SD): **49.8 (9.8)** vs. **51.1 (9.5)**,***p* = 0.020**
**Roth**, 2018 [17]	**1. Depressive Symptoms** (CES-D, range 0 = best, 12 = worst, cut-off for depression ≥4)	**1. Depressive symptoms**, mean (SD): **1.4 (2.3)** vs. **1.0 (1.9)**,***p* < 0.001**
	**2. Stress** (Cohen’s Perceived Stress Scale 4-items, range 0 = best, 16 = worst)	**2. Perceived stress levels**, mean (SD): **3.6 (3.1)** vs. **3.2 (2.9)**,***p* < 0.001**
	**3. Hypertension** (self reported)	**3. Hypertension:** 57% vs. 58%, *p* = 0.467
	**4. Diabetes** (self-reported)	**4. Diabetes:** 21% vs. 22%, *p* = 0.141
	**5. Cardiovascular disease** (self-reported)	**5. Cardiovascular diseases: 18.7%** vs. **23.2%**,***p* < 0.001**
**Trevino**, 2018 [26]	**Major depressive episode** ([MDE] DSM-IV) **Generalized anxiety disorder** ([GAD] DSM-IV)	**Odds ratio [OR]**, **(95% confidence interval)** Past MDE, *n* (%): 85 (16%) vs. 1607 (17%), OR: 0.9 (0.7–1.1), *p* = 0.348 Current MDE, *n* (%): **22 (4.1%)** vs. **239 (2.6%)**, **OR: 1.6 (1.0–2.5)**,***p* = 0.037** Current GAD, *n* (%): **21 (3.9%)** vs. **125 (1.3%)**, **OR: 3.0 (1.9–4.8)**,***p* < 0.001** Current Comorbid MDE and GAD, n (%): **6 (1.1%)** vs. **42 (0.5%)**, **OR: 2.5 (1.1–5.9)**,***p* = 0.038** IC without past MDE: **OR: 7.7 (3.5–17.0)**,***p* < 0.001** IC with past MDE: OR: 1.1 (0.6–2.1), *p* = 0.662 Past MDE and NC: **OR: 60.3 (38.0–95.6)**,***p* < 0.001** Past MDE and IC: **OR: 8.9 (3.7–21.7)**,***p* < 0.001**
**Hong**, 2017 [27]	**Self-reported diagnoses** (depression, insomnia, hypertension, pain, diabetes)	**Prevalences of self-reported diagnoses (after matching for age**, **sex**, **education etc.):** Depression, % (*n*): **4.9 (192)** vs. **3.5 (138)**,***p* < 0.001** Hypertension, % (*n*): 33.2 (1287) vs. 32.3 (1252), *p* = 0.39 Diabetes, % (*n*): 13.8 (535) vs. 13.2 (511), *p* = 0.42 Dyslipidaemia, % (*n*): 14.2 (551) vs. 13.7 (531), *p* = 0.51 Angina pectoris, % (*n*): 3.2 (126) vs. 2.9 (114), *p* = 0.43 Heart attack, % (*n*): 2.4 (95) vs. 2.0 (80), *p* = 0.25 Arthritis, % (*n*): 20.5 (794) vs. 19.8 (767), *p* = 0.44 Osteoporosis, % (*n*): 12.6 (490) vs. 11.8 (459), *p* = 0.28 Cataract, % (*n*): 16.7 (648) vs. 16.7 (648), *p* = 1.0
**Goren**, 2016 [28]	**1. Depressive symptoms** (PHQ-9, range 0 = best, 27 = worst, cut-off for depression ≥ 10)	1. PHQ-9 mean [SD]: **4.4 [5.5]** vs. **3.2 [4.8]**,***p* < 0.05** 1. PHQ-9 ≥ 10 (MDD) % (*n*): **14.2 (185)** vs. **8.9 (4801)**,***p* < 0.05**
	**2. Self-reported diagnoses** (depression, insomnia, hypertension, pain, diabetes)	2. Depression: diagnosed % (*n*): **6.2 (81)** vs. **3.3 (1778)**,***p* < 0.05** 2. Insomnia: diagnosed % (*n*): **9.8 (128)** vs. **4.4 (2361)**,***p* < 0.05** 2. Anxiety: diagnosed % (*n*): **2.0 (26)** vs. **0.8 (448)**,***p* < 0.05** 2. Hypertension: diagnosed % (*n*): **17.5 (228)** vs. **11.7 (6290)**,***p* < 0.05** 2. Pain: diagnosed % (*n*): **15.5 (202)** vs. **7.9 (4269)**,***p* < 0.05** 2. Diabetes: diagnosed % (n): **6.1 (79)** *vs.* **3.7 (1981)**, **p < 0.05**
	**3. Quality of Life** (SF-36v2: mental and physical component summary [MCS, PCS], range 0 = worst, 100 = best. SF-6D: range 0.29 = worst, 1 = best)	3. PCS mean [SD]: **51.6 [6.6]** vs. **53.6 [6.1]**,***p* < 0.05** 3. MCS mean [SD]: **46.0 [10.7)** vs. **48.0 [9.6]**,***p* < 0.05** 3. SF-6D: mean [SD]: **0.7 [0.1]** *vs.* **0.8 [0.1]**, **p < 0.05**
	**4. Productivity impairment** (WPAI)	4. Absenteeism: % work missed mean [SD): **5.8 (15.8)** vs. **2.9 (12.4)**,***p* < 0.05** 4. Presenteeism: % impairment at work mean [SD): **22.8 (25.4)** vs. **18.6 (23.2)**,***p* < 0.05** 4. Overall work impairment in hours mean (SD): **25.7 (28.2)** vs. **20.3 (25.2)**,***p* < 0.05** 4. Activity impairment in hours mean (SD): **25.4 (25.8)** *vs.* **20.7 (24.4)**, **p < 0.05**
	**5. Self-reported healthcare resource utilization**	5. Emergency room visits (past 6 months) mean (SD): **0.3 (1.8)** vs. **0.1 (0.9)**,***p* < 0.05** 5. Hospitalizations, past 6 months mean (SD): **0.8 (5.2)** vs. **0.5 (4.1)**,***p* < 0.05** 5. Healthcare provider visits, past 6 months mean (SD):**7.7 (18.5)** *vs.* **4.4 (7.7)**, **p < 0.05**
**Laks**, 2016 [29]	**1.Depressive symptoms** (PHQ-9, range 0 = best, 27 = worst, cut-off for depression ≥10)	1. PHQ-9 mean (SD): **7.3 (7.0)** vs. **5.5 (6.0)**,***p* < 0.05** 1. PHQ-9 ≥ 10 (MDD) % (*n*): **28.7 (60)** vs. **20.4 (2176)**,***p* < 0.05**
	**2. Self-reported diagnose** (depression, insomnia, hypertension, pain, diabetes)	2. Depression: diagnosed (**OR:2.0**) % (*n*): **23.0 (48)** vs. **10.9 (1157)**,***p* < 0.05** 2. Insomnia: diagnosed (**OR:1.6**) % (*n*): **26.8 (56)** vs. **15.4 (1635)**,***p* = 0.003** 2. Anxiety: diagnosed (**OR:1.7**) % (*n*): **30.6 (64)** vs. **17.6 (1878)**,***p* = 0.001** 2. Hypertension: diagnosed (**OR: 1.6**) % (*n*): **23.4 (49)** vs. **14.5 (1547)**,***p* = 0.009** 2. Pain: diagnosed (**OR:1.7**) % (*n*): **31.1 (65)** vs. **19.0 (2020)**,***p* = 0.001** 2. Diabetes: diagnosed (**OR:2.1**) % (*n*): **12.0 (25)** vs. **4.9 (526)**,***p* = 0.004**
	**3. Quality of Life** (SF-36v2: mental and physical component summary [MCS, PCS], range 0 = worst, 100 = best. SF-6D, range 0.29 = worst, 1 = best)	3. PCS mean (SD): **51.0 (7.8)** vs. **52.2 (7.7)**,***p* < 0.05** 3. MCS mean (SD): **44.8 (12.24)** vs. **47.2 (11.14)**,***p* < 0.05** 3. SF-6D mean (SD): **0.68 (0.139)** vs. **0.72 (0.137)**,***p* < 0.05**
	**4. Productivity impairment** (WPAI)	4. Absenteeism: % work missed mean (SD): **10.1 (19.55)** vs. **6.1 (16.88)**,***p* < 0.05** 4. Presenteeism: % impairment mean (SD): **26.6 (31.60)** vs. **16.8 (25.26)**,***p* < 0.05** 4. Overall work impairment (hours) mean (SD): **30.8 (33.47)** vs. **20.3 (28.66)**,***p* < 0.05** 4. Activity impairment in hours mean (SD): **26.8 (29.85)** vs. **20.9 (27.84)**,***p* < 0.05**
	**5. Self-reported healthcare resource utilization**	5. Emergency room visits, past 6 months mean (SD): **0.8 (1.93)** vs. **0.5 (1.74)**,***p* < 0.05** 5. Hospitalizations, past 6 months mean (SD): **0.4 (2.24)** vs. **0.2 (0.94)**,***p* < 0.05** 5. Healthcare provider visits, past 6 months mean (SD): **6.6 (8.30)** *vs.* **4.6 (6.54)**, **p < 0.05**
**Berglund**, 2015 [30]	**1. Self-reported long-term illness** (“Do you have any long-term illness, problems following an accident, any disability or other long-term health problem?”)	1. Yes **42.9%** vs. **36.4%**,***p* ≤ 0.01**
	**2. Self-rated health** (“How do you rate your general state of health?”)	2. Poor/very poor: **7.3%** vs. **5.8%**,***p* ≤ 0.01** 2. Neither good nor poor: **27.3%** vs. **22.5%**,***p* ≤ 0.01** 2. Good/very good: **65.5%** vs. **71.7%**,***p* ≤ 0.01**
	**3. Health-related quality of life** (CDC HRQOL-4)	3. Days with poor physical health (last 30 days) mean (SD): **7.3 (9.4)** vs. **6.4 (9.4)**,***p* ≤ 0.01** 3. Days with poor mental health (last 30 days) mean (SD): **6.3 (9.0)** vs. **5.3 (8.3)**,***p* ≤ 0.01** 3. Days without work capacity (last 30 days) mean (SD): **4.8 (8.9)** vs. **4.1 (8.4)**,***p* ≤ 0.01**
	**4. Psychological wellbeing** (GHQ-12: range 0 = best, 36 = worst, cut-off ≥ 12)	4. GHQ12 MD mean (SD): **9.1 (5.3)** vs. **8.9 (4.8)**,***p* ≤ 0.01** 4. Good psychological wellbeing: **78.2%** vs. **82.4%**,***p* ≤ 0.01** 4. Poor psychological wellbeing: **21.8%** *vs.* **17.6 %**, **p ≤ 0.01**
**Gupta**, 2015 [31]	**1. Quality of Life** (SF-36v2: MCS, PCS: range 0 = worst, 100 = best. SF-6D: range 0.29 = worst, 1 = best)	1. MCS mean (SD): **40.3 (10.8)** vs. **45.9 (10.9)**,***p* < 0.001** 1. PCS mean (SD): **46.8 (10.2)** vs. **49.0 (9.8)**,***p* < 0.001** 1. SF-6D mean (SD): **0.6 (0.1)** vs. **0.7 (0.1)**,***p* < 0.001**
	**2. Depressive Symptoms ** (PHQ-9: 0–4 = minimal, 5–9 = mild, 10–14 = moderate, 15–19 = moderately severe, 20–27 = severe)	2. Minimal (%): **19.9** vs. **38.6**,***p* < 0.001** 2. Mild (%): **21.1** vs. **17.2**,***p* < 0.001** 2. Moderate (%): **11.8** vs. **7.4**,***p* < 0.001** 2. Moderately severe (%):**6.5** vs. **3.8**,***p* < 0.001** 2. Severe (%): **6.5** vs. **1.6**, **p < 0.001**
	**3. Self-reported current medication use for depression**	3. Medication use (%): **17.6** vs. **8.2**, **p < 0.001**
	**4. Self-reported comorbidities** (“Have you experienced the following in the past twelve months”)	4. Narcolepsy (%): 1.2 vs. 0.5, *p* = 0.072 4. Insomnia (%): **32.4** vs. **18.5**, ***p* < 0.001** 4. Sleep difficulties (%): **42.7** vs. **28.5**, ***p* < 0.001** 4. Pain (%): **39.7** vs. **30.4**, ***p* = 0.001** 4. Anxiety (%):**37.9** vs. **23.6**, ***p* < 0.001** 4. Depression (%): **29.4** vs. **19.4**, ***p* < 0.001** 4. Heartburn (%): **31.7** vs. **22.9**, ***p* = 0.001** 4. Migraines (%): 26.6 vs. 22.4, *p* = 0.102 4. Headaches (%):**48.0** vs. **42.0**, **p = 0.048**
**Rosso**, 2015 [20]	**1. Self-reported diagnosis** (chronic disorders)	High- (≥3 x/wk) vs. Low-Frequency IC (≤2 x/wk) vs. NC: 1. Diabetes % (*n*): 7.0 (53) vs. **2.5 (35)** vs. 6.9 (241), ***p* < 0.001** (comparing high vs. low/NC) 1. Asthma % (*n*): **11.0 (82)** vs. **5.4 (73)** vs. 6.7 (231), ***p* < 0.001** (comparing high vs. low/NC) 1. Osteoporosis, % (*n*): **7.8 (58)** vs. **7.0 (95)** vs. 10.3 (353), ***p* < 0.001** (comparing high vs. low/NC)
	**2. BMI (**calculated by measured height and weight, >29.9 = obese)	2. Obese % (n):**40.4 (303)** vs. **28.9 (395)** vs. 27.5 (957), **p < 0.001** (comparing high vs. low/NC)
**Tuithof**, 2015 [32]	**1. Self-reported chronic physical disorders** (standard checklist assessed presence of 17 chronic physical disorders)	1. 45.5% vs. 37.1%, OR: 1.09, 95% CI: 0.93–1.28
	**2. Emotional Disorder** (12-month prevalence) (DSM-IV)	2. 7.5% vs. 8.8%, OR: 0.86, 95% CI: 0.66–1.11, *p* = 0.15 When informal caregiving was defined more strictly: caregiving for >8 h/wk: OR = 0.92, 95% CI: 0.64–1.31 longer than 1 year: OR = 1.22, 95% CI: 0.92–1.63 caregiving for >8 h/wk and longer than 1 year: OR = 1.17, 95% CI: 0.73–1.87
**Verbakel**, 2014 [33]	**Subjective well-being ** (self-reported happiness on a scale from 0 to 10)	Unadjusted mean (SD): 7.69 (1.71) vs. 7.66 (1.77), not statistically significant. IC had on average a slightly lower level of well-being compared to NC (−0.11, SD 0.16) that was stat. sign. after adjusting for age, sex, educational level, partner, children, co-residing parents aged 65+ and religiosity. This Well-being-difference varied across countries: in most European countries, IC reported lower levels of well-being than NC did, whereas in Scandinavia they were slightly higher in IC vs. NC. Resources of formal long-term care reduced this gap, services directed at psychosocial support, facilitating the combination of work and care and financial support did not reduce negative effects of informal caregiving.
**Chan**, 2013 [34]	**1. Depressive symptoms** (CES-D-11, range 0 = best, 19 = worst, cut-off for depression ≥ 7)	1. CES-D, mean (SD): **3.8 (3.2)** vs. **2.9 (2.6)**,***p* < 0.0001** 1. Clinically significant depressive symptoms (CESD ≥ 7): **18.2%** vs. **7.9%**, ***p* < 0.0001** **OR: 2.36**, **95% CI: 1.44–3.86**
	**2. Self-rated health** (“In general would you say your health is—excellent/very good/good/fair/poor?”)	2. Poor: **3.0%**, fair: **21.5%**, good: **57.5 %**, very good: **14.7%**, excellent: **3.4%** vs poor: **1.3%**, fair: **15.1%**, good: **64.2%**, very good: **17.0 %**, excellent: **2.5%**, ***p* = 0.02** IC were significantly more likely to have poorer SRH **OR: 2.45**, **95% CI: 1.84–3.26**
	**3. Outpatient visits** (assessed by asking if they had seen a doctor in a clinic in the last month and the number of visits)	3. No outpatient visits in the last month: 62.9% vs. 58.5%, *p* = 0.15 3. Mean number (SD) of outpatient visits (last month): 1.2 (0.8) vs. 1.2 (0.5), p = 0.42
**Herrera**, 2013 [35]	**Depressive symptoms ** (CES-D, range 0 = best, 60 = worst, cut-off for depression ≥ 16)	**Depressive symptoms**: CES-D, mean (SD): 8.0 (0.87) vs. 8.3 (0.22) CES-D ≥ 16: 14.1% vs. not reported
**Ramsay**, 2013 [23]	**Limiting long-term illness** (self-reported health problems or disability including problems that are due to age)	**Limiting long-term illness:** heavy caregiver (≥20 h/week): 32.1% light caregiver (1–19 h/week): 19.0% non-caregiver: 18.9%
**Hernandez**, 2010 [36]	**1. Depressive symptoms** (CES-D, range 0= best, 60= worst, cut-off for depression ≥ 16)	1. CES-D, mean (SD): **10.32** (10.60) vs. **6.13** (6.77), ***p* = 0.014** 1. CES-D ≥16: **24%** vs. **7%**, ***p* = 0.004**
	**2. Self-reported health ** (“How would you rate your overall health? 1 (excellent) to 4 (poor)”)	2. Mean (SD): **2.68 (0.76)** vs. **2.70 (0.75)**
**Butterworth**, 2010 [37]	**1. Anxiety and Depression** (Goldberg anxiety and depression scale)	1. Anxiety, clinically significant: **25.9%** vs. **17.5%**, ***p* = 0.003** **OR: 1.57**, **95% CI: 1.11–2.20** 1. Depression, clinically significant: **50.5%** vs. **39.3%**, ***p* = 0.002** **OR: 1.52**, **95% CI: 1.15–2.03**
	**2. Physical impairment** (range 0 = worst, 100 = best)	2. SF-12 RAND scoring method with scores < 40: **23.3%** vs. **17.5%**, **p = 0.038**

aHR = adjusted hazard ratio; vs. = versus; wk = week; SD = standard deviation; SF-12 (or 36) = Short Form Survey 12 (or 36) Item; SF-6D = Short-Form Six-Dimension; SD = standard deviation; CES-D = Center for Epidemiologic Studies Depression Scale; DSM-IV = Diagnostic and Statistical Manual of Mental Disorders, Fourth Edition, by the American Psychiatric Association; PHQ-9 = Patient Health Questionnaire 9-item; CDC HRQOL-4 = The Centers for Disease Control and Prevention’s health-related quality of life 4-item; GHQ-12 = General Health Questionnaire 12-item; MCS and PCS = Mental Component Scale and Physical Component Scale; BMI = body mass index.

## Data Availability

Not applicable since we only evaluated data that have already been published.

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
