# Peer review of "Mortality, Morbidity and Health-Related Outcomes in Informal Caregivers Compared to Non-Caregivers: A Systematic Review"

_ijerph, 2022, doi:10.3390/ijerph19105864_

Round 1

Reviewer 1 Report

The authors from performed a systematic review of recent literature on informal caregiver burden. Their review included a total of 22 studies (13 cross-sectional assessments and 9 with longitudinal follow-up). Mortality benefit was reported in some of the studies with longitudinal study design. On the other hand, morbidity burden due to conditions, such as, depression, anxiety, pain were reported to be higher among caregivers in studies with cross-sectional study design.

Major comments:

  1. The design, execution, summarization, and presentation of review findings was nicely done.

  1. Caregiver/caretaker burden is a topic on which there have been numerous publications. In fact, there are scores of papers on meta-analyses/reviews of various interventions to reduce caregiver burden i.e. healthcare community is cognizant of excess burden in caregivers:
    1. In this context, the authors need to expand on the reason for conducting this review, specifically what gaps in existing literature do the findings from such a review fill?
    2. A quick review and comparison of findings of other reviews on caregiver burden would be helpful.

  1. The authors in the abstract and discussion sections report that findings from both longitudinal and cross-sectional studies have shown excess morbidity burden among informal caregivers compared to non-caregivers. However such a conclusion appears misplaced. Findings from ref 14. Rothgang et al. alone seem to support the authors’ line and hence the inferences drawn might need re-phrasing.

  1. The conclusion regarding excess morbidity burden among caregivers is driven predominantly by findings from cross-sectional studies. With all the limitations of a cross-sectional study design (temporality/bi-directionality) to answer such research question should the authors have included these studies in their review?

Minor comments:

  1. Line 28-29: Since, this is not a meta-analyses (no pooled analyses) but mere review of findings from individual studies stating “data from 2,016,635 participants” can be avoided.

Author Response

REVIEWER I

The authors from performed a systematic review of recent literature on informal caregiver burden. Their review included a total of 22 studies (13 cross-sectional assessments and 9 with longitudinal follow-up). Mortality benefit was reported in some of the studies with longitudinal study design. On the other hand, morbidity burden due to conditions, such as, depression, anxiety, pain were reported to be higher among caregivers in studies with cross-sectional study design.

Major comments:

  1. The design, execution, summarization, and presentation of review findings was nicely done.

Response: We thank the reviewer for this overall positive comment.

  1. Caregiver/caretaker burden is a topic on which there have been numerous publications. In fact, there are scores of papers on meta-analyses/reviews of various interventions to reduce caregiver burden i.e. healthcare community is cognizant of excess burden in caregivers:

2.1 In this context, the authors need to expand on the reason for conducting this review, specifically what gaps in existing literature do the findings from such a review fill?

Response: We agree with the reviewer that there are a number of publications and some reviews on informal caregiving are already existing. However, the methodological quality and design of many previously published single studies seem suboptimal and most previous systematic reviews have been rather old (except Bom et al, 2019). Furthermore, most reviews were not as comprehensive as our approach where we aimed to include all relevant health outcomes from the informal caregiver’s perspective, including a broad spectrum of chronic diseases. Since new studies on informal caregiving are regularly being published, updates of systematic searches and stringent evaluations are needed on a regular basis to further advance our synthesized knowledge including (hopefully) studies with better quality.

As suggested by the reviewer we expanded on the reasons for conducting this systematic review in the revised manuscript and on the knowledge gaps that we aimed to tackle. Please see the introduction section of the revised manuscript, in particular the following lines:

Lines 70-79 (clean manuscript): “However, previous analyses used often cross-sectional designs where it is not possible to determine temporal relationships to estimate incidence of disease. They were often lacking an appropriate comparison group (i.e., non-caregivers from population-based samples), and showed inconsistent findings [5,7–9]. Systematic reviews with and without meta-analyses have been conducted more than 15 years ago, and included studies with nonrepresentative samples [6,12]. The most recent systematic review that we identified included studies until April 2017 and only those that were published in English. Furthermore, that review did not assess mortality as an outcome. Due to a more narrowly defined inclusion criteria they merely screened 666 records after eliminating duplicates compared to the 5,513 that we screened [13].”

Lines 82 ff.: “[…] and to synthesize the results by grouping the outcomes visually differentiated by investigation (longitudinal or cross-sectional).”

2.2 A quick review and comparison of findings of other reviews on caregiver burden would be helpful.

            Response: Following the reviewer’s suggestion, we restructured the discussion and added a new paragraph. This additional paragraph helps to compare our results with previous.

            Line 322-344: „4.2. Interpretation of the results

4.2.1 Comparison with other systematic reviews

      Although previous systematic reviews and meta-analyses included important study designs, longitudinal and cross-sectional studies, the authors did not strictly differentiate their results and conclusions taking the considerable differences of the design into account [6,12,13]. The previous reviews and meta-analyses examined mental and physical chronic disorders of informal caregivers compared to non-caregivers but did not examine mortality of the informal caregivers [6,12,13]. Mortality, as the most relevant endpoint in longitudinal studies should be included to assess the full impact of a potentially fatal exposure that has been associated with depression, cardiovascular and other relevant chronic morbidity [38]. Previous reviews suggested a negative impact of informal caregiving on mental health such as an increased risk of depression and stress, whereas the effects on physical health were smaller and less consistent [6,12,13]. With an updated literature search and stringent study quality assessments our systematic review seems to confirm previous results in terms of chronic morbidity. However, the limited longitudinal evidence due to a lack of good quality cohort studies does not allow to make strong conclusions on temporal relationships with informal caregiving and most chronic diseases. In terms of mortality, our systematic review found five longitudinal studies showing that informal caregivers had a lower mortality risk compared to non-caregivers. We found no study, which found an increased mortality risk. Although this is strong evidence, the identified studies cannot explain the mechanism of this association. These and other potentially positive aspects of informal caregiving deserve more attention by health scientists and researchers in order to draw a broader picture.”

  1. The authors in the abstract and discussion sections report that findings from both longitudinal and cross-sectional studies have shown excess morbidity burden among informal caregivers compared to non-caregivers. However such a conclusion appears misplaced. Findings from ref 14. Rothgang et al. alone seem to support the authors’ line and hence the inferences drawn might need re-phrasing.

Response: Following the reviewer’s suggestion, we rephrased the conclusion (Abstract) and conclusion (main text) of the revised manuscript. In the clean version, please see lines 30-40 (Abstract):

            “Regarding chronic morbidity outcomes, the results from a large longitudinal German health insurance evaluation showed increased and statistically significant incidences of severe stress, adjustment disorders, depression, diseases of the spine and pain conditions among informal caregivers compared to non-caregivers. In cross-sectional evaluations, informal caregiving seemed to be associated with a higher occurrence of depression and of anxiety (ranging from 4%-51% and 2%-38%, respectively), pain, hypertension, diabetes and reduced quality of life. Results from our systematic review suggest that informal caregiving may be associated with several mental and physical disorders. However, these results need to be interpreted with caution, as the cross-sectional studies cannot determine temporal relationships. The lower mortality rates compared to non-caregivers may be due to a healthy carer bias in longitudinal observational studies, however, these and other potential benefits of informal caregiving deserve further attention by researchers.”

and lines 543-554 in main text (Conclusion):

            “It seemed that informal caregivers had a lower mortality risk compared to non-caregivers. A healthy carer bias in longitudinal population-based studies may have contributed to this finding, but informal caregiving may have positive causal effects for the health of family caregivers. These potential benefits require further research.

In terms of chronic morbidity, our systematic review showed statistical associations of informal caregiving with the development of severe stress and adjustment disorders, depression, anxiety, sleep disorders, diseases of the spine and back, pain conditions and a lower quality of life. These effects seemed stronger among informal caregivers who cared for dementia or schizophrenia patients. However, these results need to be interpreted with caution, as most of the included studies had a cross-sectional design that  does not allow to determine temporal or causal relationships.

  1. The conclusion regarding excess morbidity burden among caregivers is driven predominantly by findings from cross-sectional studies. With all the limitations of a cross-sectional study design (temporality/bi-directionality) to answer such research question should the authors have included these studies in their review?

Response: We agree with the reviewer about the limitations of cross-sectional studies. As mentioned in our response to comment no. 3 of the reviewer, we therefore rephrased the conclusions in the revised manuscript. However, we did not want to exclude cross-sectional studies, since the number of longitudinal studies, especially with good methodological quality, is rather low in this research field. By describing the statistical associations in population-based cross-sectional evaluations, we aimed to contribute to the field by not only describing the occurrence and showing what has and what has not been examined and thus also to generate ideas for further longitudinal research, which is strongly needed. We therefore would like to keep these studies included in the systematic review and hope to have rephrased he conclusions more adequately and carefully.   

Minor comments:

  1. Line 28-29: Since, this is not a meta-analyses (no pooled analyses) but mere review of findings from individual studies stating “data from 2,016,635 participants” can be avoided.

Response: We followed the reviewer suggestion, deleted the statement and shortened the corresponding sentence. Please see line 28 f in the revised (clean) manuscript: “We included 22 studies, which came predominately from the USA and Europe.”

Reviewer 2 Report

Thank you for the opportunity to review this manuscript. This systematic review is with the focus on informal caregivers’ vs non-caregivers’ mortality and mental and physical morbidity combining longitudinal and cross-sectional studies. It is a huge mission with many pitfalls most of them recognized in the discussion section (limitations). One of them not recognized is the impact from the country differences in the health care system, i.e. if informal caregiving is part of the welfare contract between citizens and the public system, a choice or not? Is it a choice or is it a duty? Another problem is that most of the research on informal caregiving is built on (often not clearly verbalized) that informal caregiving is negative for the caregiver and that assumption has implications for the methods and assessments made, commonly only assessing negative outcomes. Thus, research is portraying only the negative side of being an informal caregiver.

The methods used, retrieving studies and quality assessment is adequate and of high quality. Having said this, it did not become clear if the longitudinal studies were longitudinal in the sense of informal caregiving and its impact on mortality and morbidity. This means that at least a baseline assessment and a follow up using the variables assessing morbidity and mortality at least at two points in time should be presented. If not,  the studies are merely cross-sectional in this respect. Also, it was not clear why the studies by Roth, Rosso and Ramsay is presented as cross-sectional if longitudinal data is available, perhaps not published data but available by contacting the authors. In addition, the studies are supposed to be drawn from the population, but it is not clear if using a health insurance data base really is population based, i.e. how do people select that particular health insurance company? May there be a selection bias and is there a risk of bias in what is reported to the health insurance company, i.e. if reporting a problem do you get more reimbursement? This needs to be addressed. In sum, the material used for the analyses is confusing and the difference between longitudinal and cross-sectional data needs to be more clearly described and perhaps also consider reporting either longitudinal (truly longitudinal) or cross-sectional data. Also make sure that it is population-based samples.

As for the result section it is very difficult to grasp the results. The sample description (table 1) column 5 and 6 sometimes overlap and sometimes do not report relevant data for instance description of care recipient is said to be spouse. Also consider adding for instance the variables related to mortality and morbidity in the table and assessed at baseline vs follow up and some information about informal caregiver at baseline, i.e. health status, length of caregiving etc. Another problem making the result section very difficult to read is the fact that the text and tables are not in line. The text reveals responses to the main questions; mortality and morbidity whilst the tables are organized in relation to studies and thus there is repetition not needed and very difficult to understand. Text and tables should follow each other. In relation to the text it would be helpful to know for instance, do all longitudinal studies report mortality or a certain aspect of morbidity or is it only a few studies reporting such information.

As for the discussion section, it is merely repetition of results and not taking the findings to a more abstract level. It is difficult to write a discussion when to material is so massive as it is in this study. One way of dealing with that is stick to the aim and focus on mortality, mental vs physical morbidity that can be related to informal caregiving. For instance, it is self-evident that if you are depressed you most commonly have anxiety and feel stressed. Also, the interpretation that low mortality in informal caregivers may be due to healthier people accepting to become caregiver is guessing. Such data is not presented in the study. Being an informal caregiver for instance, as a spouse is much more complicated. Since this study and so many other studies fail to explore also the positive sides of providing informal care other aspects that could be supportive remains hidden. In essence, the bigger picture of the findings needs to be addressed in the discussion section.  

Author Response

REVIEWER II

  1. Thank you for the opportunity to review this manuscript. This systematic review is with the focus on informal caregivers’ vs non-caregivers’ mortality and mental and physical morbidity combining longitudinal and cross-sectional studies. It is a huge mission with many pitfalls most of them recognized in the discussion section (limitations).

One of them not recognized is the impact from the country differences in the health care system, i.e. if informal caregiving is part of the welfare contract between citizens and the public system, a choice or not? Is it a choice or is it a duty?

            Response: We thank the reviewer for pointing this out. It is certainly another limitation that needs to be recognized. We therefore added this aspect in the Discussion section under potential limitations in the revised manuscript (lines 523-540).

            “Sixth, differences in health care systems may contribute to different health outcomes of family carers. In many countries, informal caregiving is part of the welfare contract between the citizens and the public social and health care system. There may be health system related and/or cultural differences on how much relatives may feel obliged to become engaged in informal caregiving. However, the studies we included in the present review did not collect data on these complex issues, which require further multidisciplinary research efforts. Seventh, additionally to population-based studies, we also included register data or health insurance database. Since some health insurance companies in Germany cover a large part of the population and membership is open to all employees, with very few exceptions, we did not want to exclude them from the review. One advantage of using routine care data (retrospectively) from large insurance databases is that they include data from persons who would usually not consent to participate in prospective research projects, thus this data may even be subject to less selection bias than primary research projects.  However, although no single health insurance company in Germany can be considered as representative for the whole country, data from the members of a large company that include information of the family caregiver status may provide an additional insight into potential associations [74].”

1.1 Another problem is that most of the research on informal caregiving is built on (often not clearly verbalized) that informal caregiving is negative for the caregiver and that assumption has implications for the methods and assessments made, commonly only assessing negative outcomes. Thus, research is portraying only the negative side of being an informal caregiver.

Response: We agree with the reviewer, as this was also our impression from the comprehensive search of population-based studies including careful review of the references of all included studies to identify further studies. Please see also our response to comment 4.2 by this reviewer.

  1. The methods used, retrieving studies and quality assessment is adequate and of high quality. Having said this, it did not become clear if the longitudinal studies were longitudinal in the sense of informal caregiving and its impact on mortality and morbidity. This means that at least a baseline assessment and a follow up using the variables assessing morbidity and mortality at least at two points in time should be presented. If not, the studies are merely cross-sectional in this respect.

Response: We included also cross-sectional evaluations of data that were collected in longitudinal studies but only at one time point. We consistently used the term “cross-sectional evaluation” and hope that this is now clearer in the revised manuscript, please see lines 141-148:

      “Relevant data were extracted from the articles and two tables were composed in order to summarise basic study information on study design, sample size, mean age and sex, differentiated between longitudinal (Table 1) and cross-sectional studies (Table 2). Table 3 (longitudinal studies) and 4 (cross-sectional evaluation) present the assessment tools and outcomes, where we allocated data that were collected only at one time point in longitudinal studies to cross-sectional evaluation.

2.1 Also, it was not clear why the studies by Roth, Rosso and Ramsay is presented as cross-sectional if longitudinal data is available, perhaps not published data but available by contacting the authors.

      Response: These were published results based on cross-sectional evaluations because the authors performed only a cross-sectional analysis, i.e., it was only assessed at one time point. We can only speculate why they did not perform longitudinal evaluations: A possible reason is that the authors changed their research focus during the long-term study and added new questions to the follow-up assessment or deleted old ones. Or their research team had changed.

We followed the reviewer’s suggestion and contacted these three authors. Until today, only Rosso answered and declined further analyses in terms of longitudinal analyses. We have not received a response from the other two authors, if unpublished results or data is available.  

2.2 In addition, the studies are supposed to be drawn from the population, but it is not clear if using a health insurance data base really is population based, i.e. how do people select that particular health insurance company? May there be a selection bias and is there a risk of bias in what is reported to the health insurance company, i.e. if reporting a problem do you get more reimbursement? This needs to be addressed.

      Response: We agree with the potential bias that the reviewer is pointing out. However since some health insurance companies (in Germany) cover a large part of the population (and membership is open to all employees, with very few exceptions), we did not want to exclude them from the review. We listed these data sources as a second inclusion criteria in addition to population-based studies, trying to avoid the term “population-based” for them. We described them under the term “register data or a health insurance database” (please see line 120 in the revised manuscript). Reporting health-related outcomes are not associated with higher reimbursement within German statutory health insurances (data source of Rothgang et al.). However, reporting or recording of diagnoses by doctors may be related to some measurement bias, which may go in both directions. We cannot rule out that this may have played a role. We added this form of measurement bias as a potential limitation. Please see lines 529-540 in the revised manuscript.

“Seventh, additionally to population-based studies, we also included register data or health insurance databases. Since some health insurance companies in Germany cover a large part of the population and membership is open to all employees, with very few exceptions, we did not want to exclude them from the review. One advantage of using routine care data (retrospectively) from large insurance databases is that they include data from persons who would usually not consent to participate in prospective research projects, thus this data may even be subject to less selection bias than research projects including original data.  However, no single health insurance company in Germany can be considered as representative for the whole country, but results from a large company may provide an additional insight into potential associations [74].”

2.3 In sum, the material used for the analyses is confusing and the difference between longitudinal and cross-sectional data needs to be more clearly described and perhaps also consider reporting either longitudinal (truly longitudinal) or cross-sectional data. Also make sure that it is population-based samples.

      Response: We followed the reviewer’s suggestion and described our allocation regarding variables and data. Please see lines 143-147.

       “[…] and sex, differentiated between longitudinal (Table 1) and cross-sectional studies (Table 2). Table 3 (longitudinal studies) and 4 (cross-sectional evaluation) present the assessment tools and outcomes. We allocated variables that were collected only at one time point (e.g. only at baseline) within longitudinal studies to cross-sectional evaluation.”

Furthermore we rearranged the structure of the discussion by separating it into longitudinal and cross sectional evaluations in the same way as the result section is structured. Please see 4.2.2 – 4.3 within the revised manuscript.

  1. As for the result section it is very difficult to grasp the results. The sample description (table 1) column 5 and 6 sometimes overlap and sometimes do not report relevant data for instance description of care recipient is said to be spouse.

      Response: Following the reviewer’s feedback we reordered Table 1 and Table 2. We combined column 5 (Definition of informal caregiver) and column 6 (Definition of care recipient) into one (Description of sample).

3.1 Also consider adding for instance the variables related to mortality and morbidity in the table and assessed at baseline vs follow up and some information about informal caregiver at baseline, i.e. health status, length of caregiving etc.

      Response: We followed the reviewer’s suggestion and added corresponding information in Table 1, for Rothgang (2018), Fredman (2015) and Kenny (2014).

3.2 Another problem making the result section very difficult to read is the fact that the text and tables are not in line. The text reveals responses to the main questions; mortality and morbidity whilst the tables are organized in relation to studies and thus there is repetition not needed and very difficult to understand. Text and tables should follow each other.

Response: We thank the reviewer for pointing out this aspect and restructured the text so that its structure now follows the structure of the tables.

The discussion is restructured from 4.2 Interpretation of the results to 4.2.3.2 Health-related outcomes.

3.3 In relation to the text it would be helpful to know for instance, do all longitudinal studies report mortality or a certain aspect of morbidity or is it only a few studies reporting such information.

      Response: Five out of nine longitudinal studies that we included reported mortality (see lines 195 clean version). The two matrices (in revised manuscript Figures 3 and 4) give an overview with respect to the type of evaluation (longitudinal or cross-sectional) and the outcomes that were examined in each of the included studies. Roth (2018), Rosso (2015) and Ramsay (2013) conducted longitudinal studies but also assessed outcomes only at one time point (for example at baseline). In order to express that these outcomes are only measured at one point, we classified them as “cross-sectional evaluations” (see figures 3 and 4 and tables 3 and 4).

Different from the overview of results/outcomes, Table 1 (longitudinal studies) and Table 2 (cross-sectional studies) present only the basic information about the design of the study, where data was collected from.

  1. As for the discussion section, it is merely repetition of results and not taking the findings to a more abstract level. It is difficult to write a discussion when to material is so massive as it is in this study. One way of dealing with that is stick to the aim and focus on mortality, mental vs physical morbidity that can be related to informal caregiving. For instance, it is self-evident that if you are depressed you most commonly have anxiety and feel stressed.

Response: We thank the reviewer for this comment and followed his/her suggestion, also considering other comments by the reviewers about the interpretation of our findings. In the revised manuscript, we restructured the discussion and also separated it into interpretation of findings from longitudinal and cross-sectional studies. Within this distinction we stick to the aim of this review and discussed the massive material in order of mortality, morbidity and health-related outcomes (4.2 – 4.3). Further, we added a new paragraph where we deepened the discussion in relation to other previously published systematic reviews (4.2.1 Comparison with other systematic reviews). In the revised (clean) manuscript please see lines 323. 

4.1 Also, the interpretation that low mortality in informal caregivers may be due to healthier people accepting to become caregiver is guessing. Such data is not presented in the study.

      Response: We agree with the reviewer that this interpretation must still be considered as speculative and other causal mechanisms than a healthy carer bias may play a role for the lower mortality. We added information of studies that underline a healthy caregiver bias, please see lines 357-362:

      “There have been some longitudinal studies that suggest that informal caregivers are healthier [40–42]: the study from Bertrand et al (2012) showed that cognitive outcomes in older women caregivers were better than in non-caregivers of the same age [40]. McCann et al 2004 showed that care-giving older adults were physically healthier than their non-caregiving counterparts, and Fredman et al (2009) showed a lower rate of functional decline among informal caregivers [41,42].”

4.2 Being an informal caregiver for instance, as a spouse is much more complicated. Since this study and so many other studies fail to explore also the positive sides of providing informal care other aspects that could be supportive remains hidden. In essence, the bigger picture of the findings needs to be addressed in the discussion section.

      Response: We thank the reviewer for pointing out that positive aspects for the informal caregiver have been understudied. We tried to address this complex issue in the revised manuscript. Please see lines 454-471:

4.3. Positive aspects of informal caregiving
      Apart from a potentially reduced mortality, our systematic review  did not identify much evidence for protective effects of informal caregiving in terms of reduced chronic morbidity, although we did not exclude these outcomes. Beyond the focus of our present work based on stringent methodological quality assessments, however, further research exists suggesting that the provision of emotional and practical support to others may result in improved mental and/or physical health for the provider of such support [62–64]. Informal caregiving can lead to a high level of self-esteem and a positive change in sense of mastery among femal informal caregivers, when caring for a non-resident care recipient [65,66]. In addition, it also seems to have a stress-buffering effect which leads to  lower mortality [17].
      Informal caregivers constitute a heterogeneous group regarding the amount of care, the caregiving situation (resident or non-resident, caring for a parent or a spouse), the disease of the care recipient and perceived social support. All these factors can contribute to experience a situation as stressful, which may be experienced as eustress (i.e., beneficial stress) but often as distress. According to the definition of Lazarus and Folkman, this occurs if a person appraises the (caregiving-)situation as taxing or exceeding the available resources [67].

Round 2

Reviewer 1 Report

  1. The authors have made appropriate changes to the text in response to review comments
  2. The study limitations have been elaborately detailed and hopefully readers will interpret study findings keeping the limitations of the included studies in mind (with regards to cross-sectional studies).

Author Response

  1. The authors have made appropriate changes to the text in response to review comments
  2. The study limitations have been elaborately detailed and hopefully readers will interpret study findings keeping the limitations of the included studies in mind (with regards to cross-sectional studies).

            Response: We thank the reviewer for the positive feedback to our revised manuscript.

Reviewer 2 Report

The authors have taken comments made into consideration. Still I think the result section could be improved by greater focus on the aims and reporting the results accordingly and not so much on reporting the results of the individual studies. 

Author Response

The authors have taken comments made into consideration. Still I think the result section could be improved by greater focus on the aims and reporting the results accordingly and not so much on reporting the results of the individual studies.

      Response: We thank the reviewer for this comment and followed his/her suggestion to put more focus on the aims of the systematic review when presenting the results. Therefore we restructured the result section from para 3.3 Longitudinal studies to para 3.4.3 Health-Related Outcomes. This reflects better the order of the aims of this review (mortality, morbidity and health-related outcomes). Furthermore we updated Figure 3 to present the results accordingly in a consistent way. We hope with this revision, we were able to implement what the reviewer meant regarding the result section.